# Cell Motility and Cancer

**DOI:** 10.3390/cancers12082177

**Published:** 2020-08-05

**Authors:** Ildefonso M. De la Fuente, José I. López

**Affiliations:** 1Department of Nutrition, CEBAS-CSIC Institute, Espinardo University Campus, 30100 Murcia, Spain; 2Department of Mathematics, Faculty of Science and Technology, University of the Basque Country, 48940 Leioa, Spain; 3Department of Pathology, Cruces University Hospital, Biocruces-Bizkaia Health Research Institute, 48903 Barakaldo, Spain

**Keywords:** cell motility, migration, conditioned behavior, learning, cancer, invasion, metastasis

## Abstract

Cell migration is an essential systemic behavior, tightly regulated, of all living cells endowed with directional motility that is involved in the major developmental stages of all complex organisms such as morphogenesis, embryogenesis, organogenesis, adult tissue remodeling, wound healing, immunological cell activities, angiogenesis, tissue repair, cell differentiation, tissue regeneration as well as in a myriad of pathological conditions. However, how cells efficiently regulate their locomotion movements is still unclear. Since migration is also a crucial issue in cancer development, the goal of this narrative is to show the connection between basic findings in cell locomotion of unicellular eukaryotic organisms and the regulatory mechanisms of cell migration necessary for tumor invasion and metastases. More specifically, the review focuses on three main issues, (i) the regulation of the locomotion system in unicellular eukaryotic organisms and human cells, (ii) how the nucleus does not significantly affect the migratory trajectories of cells in two-dimension (2D) surfaces and (iii) the conditioned behavior detected in single cells as a primitive form of learning and adaptation to different contexts during cell migration. New findings in the control of cell motility both in unicellular organisms and mammalian cells open up a new framework in the understanding of the complex processes involved in systemic cellular locomotion and adaptation of a wide spectrum of diseases with high impact in the society such as cancer.

## 1. Introduction

Cell migration is essential for a plethora of fundamental biological processes and human pathologies such as cancer. The molecular and biochemical mechanisms through which individual cells move have been extensively studied. However, the principles that govern cell motility at a systemic level are still largely unknown. This narrative reviews seminal aspects of cell motility and its application to cancer, in particular, the usefulness of analyzing systemic properties of unicellular eukaryotic organisms to understand cancer cell migration. The importance of a systemic approach to the external stimuli involved in cellular locomotion have provided important findings, such as the limited role of the nucleus in cell motility on two-dimension (2D) surfaces and the emergence of a new behavior by which the cells do learn and develop an associative memory to respond to the environmental changes during cell migration.

Cancer is a health problem of major concern and a leading cause of death in Western societies. Local invasion and metastatic seed in distant territories are complex biological processes that impact negatively in patient prognosis. These issues are receiving much attention in the last years [1]. A better knowledge of the systemic mechanisms underlying cell motility is necessary to advance in the development of efficient therapies to improve cancer prognosis.

## 2. Simple Organism Models Are Necessary to Understand Human Cell Behavior

Locomotion is a crucial ability to survive for many unicellular eukaryotic cells and the translation of the information obtained from these single organisms to human cancer and other diseases is a milestone in modern medicine. In general terms, motility -understood as cell displacement- has been well conserved over hundreds of millions of years of evolution from unicellular eukaryotic organisms to human cells. For this reason, many studies have analyzed cell motility in non-mammalian model organisms, like amoebae, worms, flies and others. Elegant studies with *Batrachochytrium dendrobatidis* [2,3,4], *Dictyostelium discoideum* [5,6,7,8], *Caenorhabditis elegans* [9,10,11] and *Drosophila melanogaster* [12,13,14,15,16], among others, have provided useful information to understand the fundamental mechanisms of cell motility. 

As it has been recently reviewed [5,6], *D. discoideum* is an important model to analyze cell locomotion, chemotaxis and many other cell characteristics, in part because this social amoeba evolves from a unicellular to a multicellular stage during its cell cycle. *D. discoideum* has a very well developed phagocytic ability and defense mechanisms against potential pathogens making this amoeba a good mimicker of macrophages and other mammalian cells with motile properties. On the other hand, several genes in its genome are homologous to some disease genes in humans [6]. This fact makes once more this microorganism a model to analyze the mechanisms of action involved in several human diseases such as inherited Parkinson’s disease [7] or cancer [8].

Basal membrane disruption is the first step in tumor invasion and has been analyzed in simple multicellular organisms like *C. elegans* [9], where the anchor cell during its larval development breaks basement membranes during morphogenesis. Another example of the crucial information obtained from this nematode and translated to cancer are the advances in the knowledge of the molecular mechanisms of apoptosis [10]. In fact, *C. elegans* has emerged as a simple animal model for systematic dissection of the molecular basis of tumorigenesis, focusing on the well-established processes of apoptosis and autophagy [11].

More complex multicellular organisms such as *D. melanogaster* show a model of metastatic potential through the development of several mutations conferring different potentialities to increase cell migration. Such studies are improving the understanding of some fundamental processes in cancer, local invasion and metastases included, for example the discovering of the Hedgehog and WNT pathways [12]. Strikingly, mammalian and *D. melanogaster* intestines share many similarities [13]. For these reasons, this fly has been also a model to understand the development of collective cell migration and metastases through epithelial-mesenchymal transition (EMT) processes driven by the transcription factor Snail [14]. Indeed, this organism has been used for the analysis of possible therapeutic routes in cancer [15,16]. Models of glioblastoma and rhabdomyosarcoma developed in *D. melanogaster* have allowed a better knowledge of the genomic alterations underlying neoplasms. The inhibition of the RET oncogenic activity to treat multiple endocrine neoplasia by newly designed chemicals [15] is another good example of the applicability of these studies in *D. melanogaster*.

## 3. The Locomotion System in Unicellular Eukaryotic Organisms and Human Cells

Cellular migration is controlled by complex molecular and metabolic networks. These networks shape an intricate interplay of multiple components amongst which the cytoskeleton, a large number of adhesion proteins, varied signaling processes and sophisticated biochemical regulatory networks are fundamentally included.

The cytoskeleton of eukaryotic cells (the main part of the locomotion system) is a dynamic structure formed by three main components—actin microfilaments, microtubules and intermediate filaments, all of them interacting in complex dynamic networks. Cell migration in amoeboid organisms is largely dependent on regulatory mechanisms of the actin cytoskeleton, in which integrins, Rac small GTPases and many post-translational modifications such as Arp2 phosphorylation, adhesion associated proteins including talin, paxillin and vinculin and numerous intracellular signaling molecules, take part modifying the systemic migratory behavior of cells [5].

Most of the spatial-temporally regulated actin dynamics in human cells, for instance leukocytes, share great similarity to amoeboid unicellular organisms [17]. However, there are some differences between them, for example, with respect to the Cdc42 protein, a component that plays a prominent role in the directed actin dynamics of leukocytes [18]. This protein is a small GTPase of the Rho family that is present in a variety of organisms, from yeasts to mammals. Cdc42 regulates the signaling pathways that control important cellular functions including cell migration, endocytosis and cell cycle progression. Recently, Cdc42 has been directly implicated in cancer progression. Actually, Cdc42 is overexpressed in lung, colorectal, breast and testicular cancer, as well as in melanoma [19]. Albeit some variations among different cell types may occur, the basic structures of the cytoskeleton are similar in many unicellular eukaryotic organisms and human cells. These similarities are supported by highly conserved gene products and numerous metabolic processes in most eukaryotic organisms endowed with directional motility [20].

Cell-autonomous polarity is also required for adequate directionality motion and optimal chemoattractant reception [21]. There are a lot of molecular processes that have been implicated in the intrinsic polarity status of cells. One of the best studied is the receptor/G protein network, which detects the external gradients (by means of the chemoattractant receptors) and transmits the external sensorial gradient to the signal transduction system. These processes amplify the directional bias and spreads the asymmetric molecular information to the cytoskeleton system which subsequently generates a protrusive force with specific polarized cell movements according to the prevailing molecular composition in the external cellular medium. A notable difference found between unicellular eukaryotic organisms and human cells is the variable richness in the repertoire of receptors and ligands controlling directed migration in different cells. For example, while only a few chemo-attractants have been identified in *D. discoideum*, human leukocytes respond to numerous molecules, as for instance PAF (Platelet-activating factor), LTB4, C5a, Interleukin-8 (IL8) and growth factors such as IGF-1, EGF, PDGF and TGF-β [22]. 

On the other hand, GTPases of the Ras superfamily act as enzymatic central mechanism to control a wide range of essential metabolic pathways such as actin cytoskeletal integrity, cell adhesion and cell migration in all eukaryotic cells endowed with motile abilities. In amoeboid unicellular organisms, as for instance in *D. discoideum,* Ras GTPase activity is directly implicated in cell locomotion and signal transduction, where it transfers the input from the receptor/G protein network to several metabolic activities, including PI3K/PIP3, Rap1, cGMP/Myosin II and TORC2/PKB pathways. In mammalian cells GTPases of the Ras superfamily also regulate cell proliferation, differentiation, migration and apoptosis. Roughly 60 types of Ras GTPases have been identified. In leukocytes, Ras GPTase has been involved in PI3K/PIP3 and MAPK processes [5]. *D. discoideum* exhibits several members of the Ras GTPases enzymes, belonging to 14 Ras family genes with 5 characterized isoforms which share similarities with mammalian H-Ras (proto-oncogene involved in the development of several types of cancer) and K-Ras (proto-oncogene involved in the Warburg effect of cancer cells) [23]. Mutations in this Ras family of proto-oncogenes are very common in human cells, being found in 20% to 30% of all tumors [24]. Ras GTPases are highly conserved between *D. discoideum* and mammalian cells and there is a basic similarity in the overall organization of the signal transduction networks in amoeboid unicellular organisms and human cells [5].

Another important cytoskeletal remodeling is the activity of PAKs enzymes which are found in all eukaryotic cells. These groups of enzymes (p21-activated kinases) are serine/threonine protein kinases effectors of the Rho family of GTPases, which are responsible in the direct regulation of cell migration, chemotaxis, cell polarity, plasticity and signaling [25]. Three PAK families of genes have been identified in *D. discoideum* and six isoforms of PAKs are expressed in human cells [25], which are implicated in a variety of processes including cytoskeletal dynamics, cell migration, cell cycle, mitosis, apoptosis, angiogenesis, tumorigenesis and metastasis [26]. In fact, PAKs are frequently up-regulated in human diseases, including various types of cancers [27]. However, it is worth noting that mammalian PAKs activities are still not completely understood. Most of our knowledge about PAK functions has been derived from approaches in unicellular eukaryotic organisms and many of these functions are similar to those seen in human cells. Such studies have demonstrated that the basic structure and functions of PAKs are conserved across practically most eukaryotic cells.

Chemotaxis and cell adhesion are also controlled by Rap1 (Ras-proximate-1 or Ras-related protein 1), another small GTPase, which acts as a molecular switch essential for effective signal transduction being involved in important cellular functions as substratum adhesion, cell motility, apoptosis, cytoskeleton remodeling, motility and intracellular vesicular transport [28]. This enzyme was originally discovered in budding yeast as a telomere-binding protein that is activated in response to a range of stimuli through a number of second-messenger molecules, such as diacylglycerol, cAMP and Ca^2+^ [29]. Rap1 is rapidly activated in response to chemoattractant stimulation regulating the cytoskeletal structure and the adhesion processes in *D. discoideum* [30]. The mechanisms by which Rap1 controls cytoskeletal reorganizations in this unicellular eukaryotic organism are still under investigation. In human cells, however, Rap1 controls cell spreading by mediating the functions of integrins and regulating cell adhesion through the interaction and regulation of adaptor proteins. Specifically, this protein is an important mediator of adhesion, polarity and migration in leukocytes [31]. In addition, Rap1 also plays many roles during cell invasion and metastasis in different human cancers [32]. Different studies have shown that Rap1 is very highly conserved in amoeboid unicellular organisms and human cells [5].

Cell motility requires a complex orchestrated spatial-temporal regulation of thousands of biomolecules which shape complex dynamic networks that are not well understood. This complexity justifies why cell migration still remains a fundamental unresolved problem in contemporary biology with crucial implications in a wide spectrum of diseases, such as cancer. Anyway, several studies performed in unicellular eukaryotic cells have shown important advances in this area and numerous investigations have shown that unicellular organisms provide an excellent experimental model system to understand the precise role of many molecules and metabolic processes in cellular migration. In fact, most of our knowledge about molecular functions on cell motility has been derived from different approaches in lower organisms and many of these functions are similar to that seen in Metazoans and in particular in human cells.

Despite the differences between lower eukaryotes and higher organisms, the fundamental networks’ architecture, as well as the principles of the systemic complex orchestration and many individual regulatory modules of cell metabolism has been remarkably conserved during evolution.

## 4. External Stimuli, Migration and Cancer

The influence of external stimuli during cell migration has been classically analyzed by physiologists. In fact, galvanotaxis and chemotaxis have been documented in unicellular eukaryotic organisms as far as more than one hundred years ago [33]. In addition, other dynamic forces of cell guidance such as haptotaxis [34,35], barotaxis [36,37], durotaxis [38,39,40,41,42,43,44,45,46], topotaxis [47,48] and plithotaxis [49,50,51] have been described latterly as additional conditioning factors of cell migration.

Galvanotaxis (electrotaxis), that is, the ability of simple organisms to predictably react in an electric field, has been an issue of research in biology for decades. *Amoeba proteus*, for instance, which has served for more than one hundred years as a cellular model to study cell migration and cytoskeletal function [52], exhibits robust galvanotaxis [53]. The molecular processes that govern cell behavior under galvanotactic conditions are not well understood. However, it is known that different mechanisms are involved in this behavior, for example, the bidirectional traffic of Ca^2+^ through the cell membranes, the sequential events of actin polymerization/depolymerization and the actomyosin contractility [54,55,56]. 

The effect of galvanotaxis has also been analyzed in tumor cells [54]. Changes in the concentration of Ca^2+^ may influence cell adhesion, a characteristic that in cancer is related to local invasion and metastases. However, it is interesting to note that cancer cells may behave very differently from their normal counterparts under similar conditions. For example, Wang et al. [57] demonstrated that human lens epithelial cells under galvanotaxis conditions migrate, as expected, to the cathode but their transformed malignant counterparts did it to the anode. Even more, this unexpected migration to the anode of malignant cells seems to be also related to the level of tumor aggressiveness. In this sense, the experiments of Frazer et al. [58] show that a highly aggressive metastatic human breast cancer cell line migrated to the anode whereas a non-metastatic human breast cancer cell line did it to the cathode. This paradoxical effect also affects the microenvironmental compartment since the directional traffic of intravasation/extravasation of tumor-infiltrating lymphocytes may be also governed by galvanotaxis [59]. These experiments and others, show that cells react in different ways depending on their malignancy and the degree of aggressiveness, opening new opportunities for cancer research in vivo. 

Chemotaxis is another essential process implicated in cell migration. For instance, chemokines-mediated chemotaxis, a phenomenon that stimulates local invasion and cell migration in malignant tumors [1], has also been previously detected in bacteria, amoebae and other unicellular eukaryotic organisms [1,5,60]. Several types of individual and collective cell migration have been described during Zebrafish morphogenesis through intercellular signaling [1]. Tumor cells at the border of infiltration display a collective behavior via E-cadherin between tumor cells themselves and with the microenvironment [61]. In this sense, macrophages, lymphocytes and neutrophils are modulated by chemokines in pituitary neuroendocrine tumors [62]. Also, tumor-associated fibroblasts increase local tumor aggressiveness favoring the development of metastases, as it has been reported in different tumors [63,64,65]. Cell migration due to chemotaxis can also be observed in processes related to tumor development, as it has been demonstrated with T-cells serving as the basis for the recently developed immune therapies [66]. In this sense, the PD-1/PD-L1 axis blockage is a promising therapeutic tool that is being successfully used in several types of malignant tumors, including kidney [67], breast [68], lung [69] and bladder [70] cancers, as well as malignant melanoma [71]. Chemotaxis can be associated to the so-called haptotaxis, which refers to the directional motility induced by a gradient of cell adhesion [34] and differs from chemotaxis in the nature of the chemoattractant molecule. So, the chemoattractant is soluble in chemotaxis and insoluble (linked to the extracellular matrix) in haptotaxis. A recent work has demonstrated fibronectin mediated haptotaxis driving directional movements in breast cancer cells during metastatic progression [35]. 

On the other hand, the capacity of migration following hydraulic gradients in the absence of chemical cues is called barotaxis. Several experiments have demonstrated that cells confined to bifurcating channels select the channel of lower hydraulic resistance to orientate and migrate [36,37]. Interestingly, barotaxis does not need chemical attractors, although both barotaxis and chemotaxis may cooperate influencing cell migration, for example in neutrophils [36] and dendritic cells [37], under specific circumstances. In a system of competition between chemotaxis with the peptide fMLP against barotaxis, Prentice-Mott et al. [36] showed that large cells with high asymmetrical capacities did not respond to chemotactic stimulus and directed their movements to low-resistance channels, whereas neutrophils (small cells with lesser asymmetric potential) could successfully overcome high hydraulic pressures to reach the chemotactic stimulus. Besides, barotaxis may guide dendritic cells to reach the lymph nodes to initiate the immune response regulating micropinocytosis by means of modification in the actomyosin cytoskeleton [37]. 

In addition, durotaxis refers to the capacity of cells to migrate following matrix stiffness cues and was originally described in fibroblasts [38]. Contact guidance, that is, the capacity of cells to follow extracellular matrix fiber orientation is the context in which durotaxis develops [39]. This cellular systemic property can be observed in many unicellular and multicellular organisms like *D. discoideum* [40] and *C. elegans* [41]. *C. elegans*, for example, is able to detect, adapt and migrate to stiffer regions in the environment by means of undulatory waves in a dorsal-ventral plane that can be modified in shape and speed depending on the changing surrounding parameters [42]. Durotaxis is involved in embryogenesis, organogenesis, inflammation, tissue repair and other physiological processes. Vascular smooth muscle cells and other mesenchymal cells, for example, show a directional motility tendency generated by increasing gradients of extracellular stromal rigidity [43,44]. Isenberg et al. [43] have provided evidence of cellular adaptations in smooth muscle cells to durotactic gradients performed on polyacrylamide gels and hypothesize with connections between chemotactic and durotactic phenomenological responses. Durotaxis appears also in cancer, for instance, tumor cell lines from glioblastoma, breast carcinoma and mesenchymal fibrosarcoma do exhibit this behavior [45]. Here, all tumor cells show a similar pattern to move towards high-stiffness gradients. The stiffening of the stroma occurring in desmoplastic tumors may offer a via of durotactic escape to cells under selective pressures, like hypoxia, for example in colorectal cancer, where extracellular matrix changes with collagen overexpression, pathological collagen crosslinking and fiber arrangement take place [46]. This way, durotaxis could contribute with chemotaxis and other external stimuli to local invasion and metastases in many tumors.

Besides, topotaxis has been recently described as the capacity of cell to mediate their migration following the density of extracellular matrix fibers [47]. Since this cell property depends of the specific stiffness of the cytoskeleton, different migratory behaviors have been observed between benign and malignant counterparts of the same cell. The loss of PTEN in aggressive variants of malignant melanoma modifies cell stiffness making them softer then switching the topotactic polarity and migration [47]. By contrast, topotaxis of melanoma cells with preserved PTEN expression migrate to areas with denser extracellular matrix making aggressive spread more difficult. A similar process has been demonstrated in cutaneous fibroblasts during wound repair [48]. 

Finally, plithotaxis explain driving forces for collective cell migration in cellular monolayers in vitro [49]. This collective cell behavior takes place in many different contexts with endothelial, fibroblasts and epithelial cells, for example, during morphogenesis of complex branched organs like lungs or kidneys, in the process of wound repair [50] and also in the collective invasion of carcinoma cells [51]. Two different cell-to-cell junction stresses working combined are main actors in the underlying mechanism governing this collective cell behavior—the normal stress or forces working perpendicular to the surface that can be tensile or compressive and the shear stress or forces applied parallel to the tangent of the surface [49]. This mechanism allows groups of cells to move collectively following chemical or physical gradients. How individual cells move inside a collectivity is explained, among other processes, by plithotaxis, an emergent dynamic property necessary to understand complex biological processes involving millions of cells like those participating in tumor invasion and metastases. 

## 5. The Role of the Nucleus in Cell Migration

One of the central issues in cellular migration is the role of the nucleus in the regulation of the locomotion system. The nucleus has been classically considered to be a key structure in cellular migration but its exact role is being understood only very recently. Graham et al. [72] have observed in human enucleated cells (cytoplasts) that the migratory abilities of these cells in 2D surfaces do not depend on the presence of the nucleus. However, cells do require the physical presence of the nucleus to move properly in three-dimensional (3D) spaces. In 3D contexts, the nucleus acts just as a necessary clutch to regulate the adaptative locomotive responses to their mechanical environment in a permanent interplay with the cytoskeleton to which this cellular structure is intimately connected [72,73].

An independent group has verified similar findings to those of Graham et al. [72] in a quantitative analysis in which the authors analyzed the movement trajectories of enucleated and non-enucleated *Amoeba proteus* sp. on flat 2D surfaces using advanced non-linear physical-mathematical tools [74] (Figure 1). The study had been previously deposited in bioRxiv.org by 2017 [75] and represents the first quantitative analysis of cell migration with enucleated cells.

To characterize the movements of cells and cytoplasts from a mathematical perspective, these authors analyzed first the relative move-step fluctuation along their migratory trajectories by applying the root mean square fluctuation (rmsf). This approach is a classical method in Statistical Mechanics based on Gibbs and Einstein’s studies [76,77] that has been latterly developed and widely applied to quantify different time-series. The obtained results showed that both cells and cytoplasts displayed migration trajectories characterized by non-trivial long-range positive correlations (Figure 1A,B). Strong correlations over periods of about 41.5 min on average were found in all the analyzed cells and cytoplasts, which corresponded to non-trivial dependencies of the past movements lasting approximately 1245 move-steps. Therefore, each cellular move-step at a given point is strongly influenced by its previous trajectory. This dynamic memory (non-trivial correlations) represents a key characteristic of the movements during cell migration. Besides, this analysis indicated that the move-step fluctuations of all amoebas presented scale-invariance properties related to the increment of the move-step length [74]. 

Next, the Mean Square Displacement (MSD) was calculated to quantify the amount of space explored over time by the amoebas and the overall migration efficiency. This method was also proposed by Albert Einstein in his work concerning Brownian motion [78]. This approach showed that the migratory trajectories of enucleated and non-enucleated amoebae cells were associated with a non-linear dependence of MSD with time, known as anomalous diffusion, which typically occurs in complex systems with long–range correlated phenomena. Therefore, a super-diffusion process governed all the efficient migration trajectories of cells and cytoplasts (Figure 1B,C). This analysis was furtherly validated by an alternative approach, the renormalization group operator (RGO) developed by Kenneth Wilson, who established the Theory of the Renormalization Group in 1971 [79]. 

Finally, to quantify some kinematic properties of the cell locomotion trajectories, the directionality ratio (DR), the average speed (AS) and the total distance traveled (TD) of the amoebas were analyzed and no significant differences were observed between cells and cytoplasts [74]. This quantitative analysis showed that both cells and cytoplasts display a kind of dynamic migration structure characterized by highly organized data sequences, non-trivial long-range positive correlations, persistent dynamics with trend-reinforcing behavior, super-diffusion and move-step fluctuations with scale-invariant properties [74].

The systemic locomotion movements of cells and cytoplasts change continuously since all trajectories display random magnitudes that vary over time. These stochastic movements shape a dynamic migration structure whose defining characteristics are preserved. Such a dynamic migration structure characterizes the mathematical way in which the locomotion movements occur and so the move-steps are efficiently organized. Since the cytoplasts preserved the dynamic properties in their migration movements similarly as intact cells, the obtained results quantitatively confirmed that the nucleus does not significantly affect the systemic movements of amoebas in 2D environments [74]. This conclusion, obtained from a mathematical and computational perspective, agrees with the results previously reported using exclusively biological techniques [72].

From a molecular point of view, the amoebas’ locomotion is controlled by complex metabolic networks, which operate as non-linear systems with dynamics far from equilibrium [80]. These biochemical networks involve an intricate interplay of multiple components of the cell migration machinery, including the actin cytoskeleton, ion channels, transporters and regulatory proteins such as the Arp2/3 complex or the ADF/cofilin family proteins [5,81]. As a consequence of the efficient self-regulatory activity of the metabolic networks, each amoeba seemed to be endowed with the ability to orientate its movement toward specific goals in the external environment, thereby developing efficient foraging strategies even in conditions of sparse resources when there is limited or no information as to where food is located. In accordance to the aforementioned studies, the enucleated amoeba’s behaviors herein observed in 2D environments may be explained by the singular self-regulated properties of the cellular metabolic life [80].

From the studies by Graham et al. [72] and de la Fuente et al. [74,75] performed in 2D surfaces, it should be taken into account that the cytoplasts generated and analyzed in these experiments are smaller than intact cell counterparts. Under these conditions, the adhesion surface is smaller and weaker in cytoplasts thus resulting in a smaller spread area, lower adhesion strength and lower total strain energy on them. In addition, the structural geometrical organization, the cytoplasmic rigidity and density and the contractility of the actomyosin cytoskeleton are also dramatically modified in the cytoplasts. Without the nucleus, the cytoplasm is very deformable and the loss of contractility of the cytoskeleton hinders the optimal travelling speed and the adequate structural functions of cytoplasts to move properly in 3D environments. 

These and other observations reveal the critical role of the nucleus for developing appropriate mechanical responses and for regulating both contractility and mechano-sensitivity [73]. Specifically, the physical presence, position and material properties of the nucleus, fundamentally those related with its connections with the cytoskeleton, are essential for a broad range of cell functions. These functions include intracellular nuclear movement, cell polarization, chromatin organization, cellular mechano-sensing and mechano-transduction signaling. Eukaryotic cells require the presence of the nucleus as a necessary component of the molecular clutch involved in the regulation of their mechanical responses to the environment. The physical properties of the nucleus strongly connected with the cytoskeleton allow and guarantee a proper cell migration when the environment displays mechanical complexities, as it happens in 3D conditions [72,73]. 

## 6. Conditioned Behavior in Single Cells

In continuation of this study and following the Pavlov’s methodological approach with dogs [82], the same group of investigators observed that two different unicellular organisms (*Amoeba proteus* sp. and *Metamoeba leningradensis* sp.) showed associative learning behaviors [83], which can be essential for adequate and efficient cellular migration (Figure 2)**.** To analyze such conditioned behavior in amoebae they used an electric field as a conditioned stimulus and a specific chemotactic peptide as a non-conditioned stimulus. The migratory trajectories of more than 700 amoebae under different experimental conditions were studied. The results showed that, through the association of stimuli, these unicellular organisms were able to learn and forget new behaviors as time passes [83]. This phenomenon can be considered as a rudimentary form of associative memory and is also crucial to govern properly cell migration.

The consecutive steps of this study [83] are summarized as follows: (1)Cell locomotion in the absence of stimuli exhibited a random directional distribution in which amoebae and metamoebae explored practically all the directions of the experimental chamber (Figure 2a),(2)Amoebae and metamoebae showed an unequivocal systemic response consisting in the migration to the cathode when they were exposed to a strong direct electric field of about 300–600 mV/mm (galvanotaxis, Figure 2b),(3)The response of both organisms was studied during biochemical guidance by exposing them to an nFMLP peptide gradient placed in the anode of a specific set-up. In these experimental conditions, most of the exposed cells migrated towards the peptide in the anode showing stochastic movements with robust directionality (chemotaxis, Figure 2c),(4)Cells were exposed simultaneously to the galvanotactic and chemotactic stimuli for 30 min (induction process). For such a purpose, the cathode was placed on the right of the set-up and the anode with the nFMLP peptide solution on the left (Figure 2d). The results showed that roughly half of the amoebae and metamoebae migrated towards the anode where the peptide was placed, while the reminders did it to the cathode,(5)To verify if the cells that moved to the anode during the induction process (Figure 2d) exhibited some degree of persistence in their migratory behavior, those cells that had previously migrated to the anode-peptide in the fourth step were exposed a second time (30 min) to a single electric field without the peptide. Under these experimental conditions, the analysis of the individual trajectories showed that most cells did migrate to the anode where the peptide was absent (Figure 2e). This evidence corroborated that a new locomotion pattern had appeared in amoebae and metamoebae (Figure 2d) (note that without the induction process practically all the cells migrated to the cathode, (Figure 2c) and after the induction process the cells modified their behavior going to the anode instead to the cathode).

This step-wise experiment showed that some amoebae seemed to associate the anode with the food (the peptide) when the cells were exposed to a stimulus related to their nourishment (the specific peptide nFMLP placed in the anode) and this exposition was simultaneously accompanied by an electric field (induction process, Figure 2d) [83]. 

After the induction process, most of the conditioned *Amoeba proteus* sp. and *Metamoeba leningradensis* sp. ran to the anode where the peptide was absent, modifying their systemic conduct, behaving against their known tendency to move to the cathode (Figure 2b) and developing a new persistent pattern of cell locomotion characterized by movements towards the anode (Figure 2e). 

Strikingly, this conditioned behavior persisted for relatively long periods ranging from 20 to 95 min (Figure 2f). The quantitative analysis of these results emphasized that it was extremely unlikely to obtain them by chance (*p* = 10^−19^; Z = 8.878, Wilcoxon rank-sum test). 

Pavlov described four fundamental types of persistent behavior provoked by two stimuli [82]. The experiment of cellular conditioning summarized here was based in one of them, the so-called *simultaneous conditioning*, in which both stimuli are applied at the same time. This finding in which individual cells can generate migratory conditioned patterns guiding their systemic locomotion movements has never been reported so far.

These experiments with unicellular organisms were the consequence of previous physical-mathematical analyses with complex metabolic networks published in 2013 [84], where using advanced tools of Statistic Mechanics and techniques of Artificial Intelligence it was verified from a computational viewpoint that metabolic networks were governed by Hopfield-like dynamics showing associative memory behavior. This quantitative study demonstrated for the first time that an associative memory was also possible in unicellular organisms. Such type of memory could be the manifestation of the emergent properties underlying the complex dynamics of the systemic metabolic networks corresponding to an epigenetic type of cellular memory [80].

## 7. Concluding Remarks

This narrative reviews fundamental issues related to cell motility and cancer. For such purpose, we revisit the usefulness of analyzing unicellular eukaryotic cells to improve current understanding in several crucial points—(i) the regulation of cell migration in mammalian cells, (ii) the importance of systemic approaches in these studies, (iii) the role of the nucleus in cell displacements, (iv) the emergence of a new behavior by which cells do learn and develop a primitive form of associative memory to respond to the environmental changes during migration and (v) the implications of some important processes related to the control of cell locomotion in cancer. The spectrum of external stimuli and the analysis of their corresponding systemic migratory behaviors (galvanotaxis, chemotaxis, haptotaxis, barotaxis, durotaxis, topotaxis and plithotaxis) in standard physiological conditions and in disease have been briefly reviewed as an example of translational research with potential clinical applicability. 

Some special points deserve further special comments, for example, the role of the nucleus in cell migration in diverse environments, fundamentally in 3D spaces. As it has been pointed out in the chapter 5 of this review, several studies have already shown that motility patterns in enucleated cells does not significantly differ from the observed in normal cells on 2D surfaces [72,74]. In this sense, the works of De la Fuente et al. [74,75] with eukaryotic unicellular organisms and of Graham et al. [72] with mammalian cells, demonstrating that cell migration does not depend on the nucleus in 2D scenarios represents a great advance in the understanding of systemic control of cell motility.

However, it is necessary to underline the crucial role of the nucleus during cell locomotion in 3D environments, for instance, cell migration during embryogenesis, wound repair and cancer is performed in 3D spaces in a context of a very different spectrum of extracellular matrix (ECM) proteins that conditionate cell movements. In this particular setting the nucleus acts as a fundamental mechanosensory structure in a continuous interplay with the cytoplasmic architecture. 

On the other hand, the comparative analysis of cell motility at the molecular level between unicellular organisms and human cells has been a matter of translational study mentioned in a specific chapter of this review. Similarities and differences have been detected in the analysis, although in general terms, crucial enzymatic routes underlying cell migration have been well preserved during evolution. Numerous studies with eukaryotic unicellular organisms have provided excellent experimental models to understand the precise role of many molecules and metabolic processes involved in cell migration. Interestingly, the translation of these findings at the molecular level from unicellular organisms to cancer cell biology has contributed to unveil new therapeutic opportunities for patients. 

Finally, the associative conditioning with long term persistence detected experimentally in unicellular eukaryotic organisms has also been highlighted. In these experiments, amoebas were able to develop a conditioned behavior during cell migration when they were exposed to two different simultaneous stimuli in a similar way as Pavlov’s dogs did more than one hundred years ago. 

The new studies in the control of cell motility both in unicellular organisms and mammalian cells define a new framework in the understanding of the mechanisms underlying the complex systemic behavior involved both in the cellular migration and in the adaptive capacity of cells to the external medium. Such findings constitute a significant advance in the comprehension of the biological processes involved in critical issues for human life like embryogenesis, tissue repair and carcinogenesis.

## Figures and Tables

**Figure 1 cancers-12-02177-f001:**
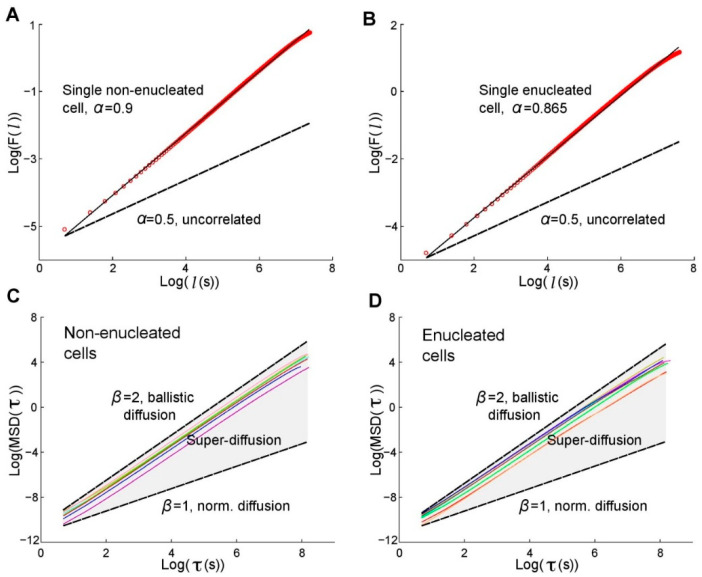
Root mean square fluctuation and Mean Square Displacement (MSD) of the trajectories of non-enucleated and enucleated amoebas. (Part of this figure has been reported previously by the authors in ref. [74].) Log-log plot of *rms* fluctuation *F* vs. *l* step for a prototype non-enucleated cell (**A**) and a prototype enucleated cell (**B**). The slope for the non-enucleated cell was *α* = 0.9, while for the enucleated it was α = 0.865, indicating non-trivial positive long-term correlations in both cases. In (**C**) and (**D**), log-log plots of MSD against the time interval τ, for 8 prototypic non-enucleated and 8 enucleated cells, respectively. β = 1 indicates normal diffusion while β = 2 indicates ballistic diffusion. The grey region defines the area of super-diffusion, in which all experimental slopes are contained. The fact that τ_max_ = 1/4th of the data length, implies that super-diffusion holds in large scales.

**Figure 2 cancers-12-02177-f002:**
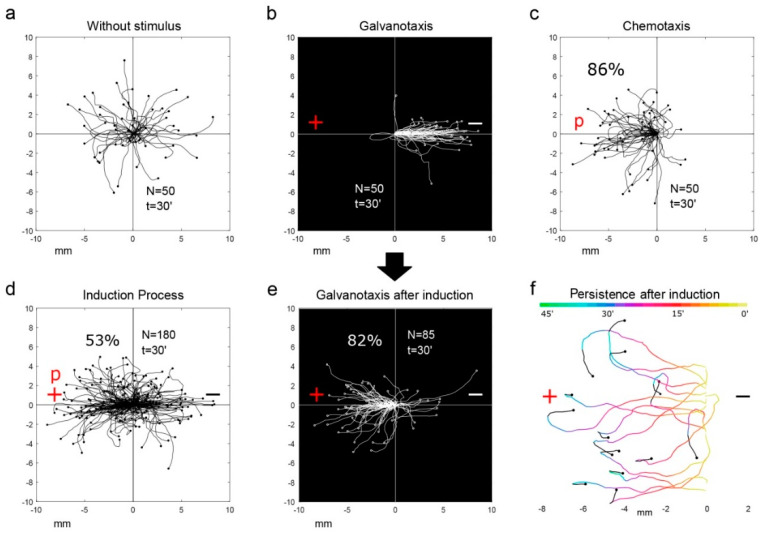
Experimental evidences of conditioned behavior in *Amoeba proteus*. (Part of this figure has been reported previously by the authors in ref. [83].) (**a**) Without stimulus, the cells practically explored all the directions of the experimentation chamber. (**b**) Under galvanotaxis, practically all the amoebae migrated towards the cathode. (**c**) Under chemotaxis, 86% of the cells migrated towards the chemotactic gradient. (**d**) Under galvanotaxis and chemotaxis simultaneously, 53% of the amoebae moved towards the anode-peptide (induced cells). (**e**) After induction process, the cells were placed in Chalkley’s medium without any stimulus for 5 min and then they were exposed to galvanotaxis for 30 min, 82% of the inducted cells presented lasting directionality towards the anode where the chemotactic peptide was absent. (**f**) Migratory trajectories of 15 amoebae under galvanotaxis, that previously acquired the conditioned behavior after induction process, lost gradually the persistence towards the anode and turned back to the cathode (times ranging from 27 to 44 min). The colors of the trajectories represent the duration of the conditioned behavior. “*N*” total number of cells, “*t*” time of galvanotaxis or chemotaxis. “*p*” chemotactic peptide (nFMLP), “+” anode, “-” cathode. In (**a**)–(**e**), both the *x* and *y* axis show the distance in mm and the initial location of each cell has been placed at the center of the diagram.

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
