# Peer review of "Cell Motility and Cancer"

_cancers, 2020, doi:10.3390/cancers12082177_

Round 1

Reviewer 1 Report

The authors provide a much improved and focused review of the field of cell migration and cancer. However, I find it unusual that they include figures directly taken from their published work, instead of providing more general views that apply to the field. In any case, it should be clear that the figures were taken from published work.

Author Response

As requeste, we have included in both legends to the figures that "part of the fugure has been previously published by the authors" and we have annotated the exact reference number. 

Reviewer 2 Report

The authors have improved the manuscript significantly. I recommend its publication, pending minor revisions:

1) The authors still cite way too many review papers. The authors need to cite original research articles.

Author Response

We agree that the manuscript includes a high number of reviews. The narrative style of the manuscript, however, claims for a general perspective when treating some topics. Anyway, we have changed 6 of them by original papers focused on more specific points (changed references are highlighted in red in the revised version of the manuscript). We hope it will be enough. 

This manuscript is a resubmission of an earlier submission. The following is a list of the peer review reports and author responses from that submission.

Round 1

Reviewer 1 Report

In this review, the authors aimed to connect the different modes of cell migration observed in simpler model organisms with migration mechanisms relevant for cancer progression. While the premise of using basic model systems as blue prints to develop a systemic understanding of cell migration is well founded, the manner by which the authors harness this concept in the review is not optimal. Indeed, the authors deviate from their goal by concurrently reviewing various and distinct concepts, such as epigenetic processes, selection, and adaptation in resistance to therapies, leaving the reader at a lost. Moreover, on several occasions throughout the review, the authors bring up specialized technical terms/concepts (such as Levy-walk strategy, robotic/soft-bodied modular robotic behavior, pavlovian-like behavior) without proper introduction or explanation. These have to be explained to allow readers from various fields/background to fully benefit from the review. Unfortunately, the overall organization of the various sections also needs attention as key concepts are often defined too late, making it very difficult to follow (see section on selection and adaptation, for example).

Overall as it stands, the review is not focused and should be rewritten with a central concept in mind.

Reviewer 2 Report

In this study, De la Fuente and Lopez provide an overview of single-cell motility and how external chemical and electrical stimuli regulate cell locomotion. Additionally, the authors discuss the role of epigenetics and selection and adaptation in cancer. In the reviewer’s opinion, the manuscript is highly underdeveloped with limited mechanistic information. The authors superficially discuss each subtopic and omit significant scientific developments.

Specific comments:

  • The authors tend to focus too much on their own work and fail to provide a current overview of the field. For example, they focus only on chemotaxis and galvanotaxis, and they do not even mention other external stimuli such as durotaxis, barotaxis which play an essential role not only in cell migration but also in decision making.
  • The authors focus only on 2D migration. It is well established that 2D does not recapitulate the complexity of the in vivo microenvironment. They need to revisit this topic and provide a comprehensive overview of the role of external stimuli in 2D, 3D, and confined migration.
  • Extensive literature exists describing the role of the nucleus in cell migration. Here, the authors primarily elaborate on their own work and conclude that the nucleus does not affect movement in 2D environments. This is certainly not incorrect, but the authors fail to provide the whole story. For example, it is known that the nucleus plays a critical role in cell migration in other microenvironments (3D, confinement). In 3D, the nucleus, which is the stiffest cell organelle, halts cell motility through very narrow pores. During cell migration, the nucleus can regulate its volume and act as a mechanosensor. Moreover, nuclear integrity can be compromised, leading to genomic instability.
  • The authors also discuss epigenetics, the Warburg effect, EMT, and resistance to therapies. Each topic can be a review paper by itself. These sections are highly underdeveloped.
  • The only exciting part of this review is related to unicellular organisms. An elaborate discussion on how unicellular organisms can be used as models for studying cell motility will be useful. How do unicellular organisms compare to mammalian cells in terms of cell locomotion (similarities and differences)?
  • The authors cite too many review papers.
  • It will be nice if the authors include a schematic and/or table summarizing this work.